# Spatial Clusters of Children with Cleft Lip and Palate and Their Association with Polluted Zones in the Monterrey Metropolitan Area

**DOI:** 10.3390/ijerph16142488

**Published:** 2019-07-12

**Authors:** Francisco Manuel Gasca-Sanchez, Jesus Santos-Guzman, Ricardo Elizondo-Dueñaz, Gerardo Manuel Mejia-Velazquez, Cecilia Ruiz-Pacheco, Deborah Reyes-Rodriguez, Elsie Vazquez-Camacho, José Ascencion Hernandez-Hernandez, Rosa del Carmen Lopez-Sanchez, Rocio Ortiz-Lopez, Daniel Olvera-Posada, Augusto Rojas-Martinez

**Affiliations:** 1Tecnologico de Monterrey, Escuela de Medicina y Ciencias de la Salud. Monterrey 64710, Mexico; 2Casa Azul, A.C. San Pedro Garza Garcia 66230, Mexico; 3Tecnologico de Monterrey, Escuela de Ingenieria y Ciencias. Monterrey 64849, Mexico

**Keywords:** cleft lip and palate, industrial pollution, PM_10_, spatial clusters

## Abstract

This study examines the spatial structure of children with cleft lip and palate (CLP) and its association with polluted areas in the Monterrey Metropolitan Area (MMA). The Nearest Neighbor Index (NNI) and the Spatial Statistical Scan (SaTScan) determined that the CLP cases are agglomerated in spatial clusters distributed in different areas of the city, some of them grouping up to 12 cases of CLP in a radius of 1.2 km. The application of the interpolation by empirical Bayesian kriging (EBK) and the inverse distance weighted (IDW) method showed that 95% of the cases have a spatial interaction with values of particulate matter (PM_10_) of more than 50 points. The study also shows that 83% of the cases interacted with around 2000 annual tons of greenhouse gases. This study may contribute to other investigations applying techniques for the identification of environmental and genetic factors possibly associated with congenital malformations and for determining the influence of contaminating substances in the incidence of these diseases, particularly CLP.

## 1. Introduction

Cleft lip and palate (CLP) is a congenital anomaly that affects the facial structure. CLP is the more prevalent congenital craniofacial anomaly worldwide, affecting between 0.7–1.5/1000 newly live births. The prevalence of CLP in Mexico has been estimated in 0.6 to 0.9/1000 births [1]. The CLP has important implications for the patient and his family, including swallowing and language development. This disorder causes psychological and social afflictions, such as discrimination, low self-esteem and difficulty to interact in society [2], as well as economic implications in terms of health care, plastic surgery, and rehabilitation. The Global Burden of Disease for CLP in 2016 was calculated at 3.4/100,000 disability-adjusted life years (DALYs) with 95% uncertainty intervals of 2.1 to 5.3 [3].

CLP is a common congenital anomaly with complex etiology, involving genetic and environmental factors [4]. Studies have identified associations between different health problems and environmental pollution [5,6,7]. On the other hand, some types of cancer, cardiovascular and respiratory diseases, and congenital malformations have been associated with different pollutants, particularly environmental particulate matter 10 micrometers or less in diameter (PM_10_), and particulate matter 2.5 micrometers or less in diameter (PM_2.5_) and the prevalence of CLP [8,9].

The Monterrey Metropolitan Area (MMA) is one of the largest cities in Mexico and Latin America, with nearly 5 million people agglomerated in 13 municipalities. This urban area is characterized by strong industrial activity and high pollution due to production of rubber and cardboard, mining of metals and stone, manufacturing of engines and industrial machinery, among others [10]. In addition to these industrial activities, the MMA concentrates more than 2 million vehicles, which exacerbate the air pollution affecting the health of the urban community. The local air-quality monitoring station (SIMA) showed the following data for the period 2009–2014: the 24-h air quality standard of PM_10_ (75 μg/m^3^) was exceeded between 224 days/year to 275 days/year and the 24-h air quality standard of PM2.5 (45 μg/m^3^) was exceeded between 21 days/year and 58 days/year. Finally, the one-hour air quality of ozone (0.095 ppm) was exceeded between 36 days/year and 95 days/year [11].

The chronic exposure to pollutants represents an important health risk for the population. The air quality data in the same period showed that the annual air quality standard of PM_10_ (40 μg/m^3^) was exceeded between 1.5 times to 2.5 times, depending on the zone in the MMA; and the annual air quality standard of PM_2.5_ (12 μg/m^3^) was exceeded two to three times. Because of this chronic population exposure, the World Health Organization qualified the MMA as one of the most polluted cities in Mexico and Latin America [12].

The population was exposed to different concentrations of pollutants across the MMA. The chemical composition of PM_2.5_ showed 50% was composed by primary components (elemental carbon, crustal material, salts, and trace metals) and secondary organic aerosols (SOA), and the other half was represented by inorganic aerosols (ammonium sulfate, ammonium nitrate) produced by different sources (refinery, industrial activity, vehicles, urban development, and wind erosion) [13]. Different health effects may be expected for these chronic exposures. Proaire 2016–2025 published a list of industrial sources of air pollutants in Monterrey and their geographical location [11].

This research aimed at answering the following research questions: do CLP cases present a random distribution or tend to concentrate in certain areas of the city? If so, what degrees of concentrations of cases are observed? Finally, what CLP cases concentrated in the space are associated with high pollution values? By answering these questions, this study may contribute to understand the epidemiology of congenital malformations in our environment. It will also identify spatial clusters over a continuous space, providing more empirical evidence in the Latin American context, since most of the studies that analyze the spatial distribution of congenital malformations are reported for discrete spaces; that is, the analysis units are usually polygons with geopolitical delimitations.

The research was structured as follows: literature review, description of investigations that identified the main risk factors for the CLP, emphasizing environmental contamination. Similarly, some studies addressing the spatial distribution of congenital malformations were reviewed. The nature and processing of the database and the spatial statistics techniques used are described below. The main results and limitations are analyzed at the end.

## 2. Environmental Risk Factors and Some Sociodemographic Characteristics of Cleft Lip and Palate (CLP)

Several reports have found a relationship between the risk of CLP and prematurity, alcohol and tobacco consumption, and drug abuse in the early stages of pregnancy [14,15,16,17,18,19]. Recently, Angulo et al. found that consumption of tobacco, the lack of vitamins and folic acid supplementation are significantly associated with CLP [20]. Environmental pollution has also been associated to CLP.

Langlois et al. found an association between CLP and radon [21]. Gonzalez et al. conducted an ecological study in Mexico and found correlations between urban environmental contamination, solid waste, life expectancy, healthcare for pregnant women and the incidence of CLP [1]. Similarly, a study by Benitez et al. [22] in Itapua, Paraguay, showed significant associations between congenital malformations and exposure to pesticides. The authors also found that pregnant women were exposed to this type of pollutants due to geographical proximity to agriculture areas where pesticides were dispersed. Garcia et al., also demonstrate an association between pesticides and congenital malformations [23].

An environmental study has documented a relationship between heavy metals exposure such as lead, nickel, mercury, cadmium, among other substances, and risk of congenital malformations such as CLP [24]. Recent studies have found evidence of the influence of environmental pollution on CLP, specifically ozone and PM_2.5_ [25]. Hwang and Jaakkola identified mothers who were exposed to air pollution during the first two months of pregnancy as having increased risk of delivering children with CLP [26]. Likewise, Desrosiers, et al. found that exposure of pregnant women to chlorinated solvents during pregnancy was positively associated with CLP [9].

Bentov et al. found a relationship between geographic proximity of industrial parks and congenital malformations [8,27]. Social exclusion could have some health implications, since groups living in marginalized areas tend to have low educational levels and poor health habits, such as smoking, drinking alcohol, being exposed to contaminants, or not taking vitamin supplements during pregnancy [28,29]. Some sociodemographic factors, such as social exclusion, low economic and educational level, and geographical marginalization have been related to increased incidence of CLP in Mexico [30]. Alfwaress et al. reported a similar situation in Jordan: CLP children were born in families with low income and low educational levels [31].

### Patterns of Spatial Distribution of Congenital Malformations: Empirical Evidence

Agay et al. [32] completed an exploratory analysis using spatial data and found that congenital malformations followed a pattern of agglomerated distribution, applying spatial autocorrelation and scan statistics techniques. Several authors have replicated these findings with different malformation in Latin America (Brazil, Argentina, Colombia) and Canada [33,34,35,36].

Efforts to detect spatial patterns of congenital malformations are well known in the literature. However, many of these studies treat space as discrete [37,38,39]; that is, territorial geopolitical units, which can be counties, municipalities and states, delimiting the analysis units, so the variable shape of study in space, is conditioned by the size and form of the territorial unit of analysis, which would tend to lead to the ecological fallacy.

## 3. Method

This was an exploratory, ecological and transversal research aimed at analyzing the spatial distribution of CLP cases and its geographical association with environmental pollution in the MMA. Although this does not establish casualty relationships, the study aims at finding spatial associations between the study groups in order to know the degree of interactions between CLP and environmental pollution.

### 3.1. Data

Clinical information was obtained from a database of patients attending Casa Azul A.C. in the last 5 years. This non-profit medical organization is dedicated to assisting low-income patients with CLP to afford integral therapy [40]. Inclusion criteria included all isolated CLP cases of 3 to 9 years, of either sex. The final sample was constituted by 333 cases, excluding syndromatic forms of CLP (Trisomies, van der Woude syndrome, Treacher Collins syndrome, etc.). Their geographical location was obtained by means of latitude and longitude coordinates. All patient families reported no urban mobility, living in the same house, at least during the patient gestational period, in order to know the exposure of the mother during pregnancy to environmental pollution interpolated to their location.

The geographical coordinates were processed using the Crimestat 3.2 and ArcGIS 10.4 software to calculate the degree of concentration and the spatial Clusters, using the Nearest Neighbor Index technique (NNI) and the Nearest Neighbor Hierarchical Clustering (NNHC) technique. With the ArcGIS software, spatial interpolation techniques were used to estimate values of environmental pollution over a continuous space, these techniques are detailed in the section on spatial statistics techniques.

For the catalog of polluting industries and their emissions, the Sistema Integral de Monitoreo Ambiental del Estado de Nuevo León (SIMA) [41] was used. This system reports the polluting substances emitted by more than 300 industries from 2010 to 2015 in Nuevo Leon. (See Table 1).

The PM_10_ data were obtained from 10 environmental monitoring stations. These data were interpolated in their mean values (see Table 2).

Likewise, a cartographic data and basic geostatistical areas shapefiles were used for the 13 municipalities that integrate the MMA. These shapefiles included data on population, education levels, health indicators, and other sociodemographic variables.

### 3.2. Spatial Statistic Techniques

#### 3.2.1. Nearest Neighbor Index (NNI) Analysis

The concentration degree, the space points, and the cluster identification were calculated for industrial emissions and the CLP cases with the NNI [42,43]. This technique compares the mean distance of the nearest points, and matches it with an expected mean distance from a random hypothetical distribution. If the mean distance is shorter than the hypothetical mean distance, it can be assumed that data follows an agglomeration pattern. On the other hand, if the difference is higher than the hypothetical mean distance, then data follows a dispersion pattern [44].

The Neighbor Nearest Distance (NND) is denoted as follows:(1)$$\mathrm{NND}\;=\;\bar{D}o/\bar{D}a$$
where: $\bar{D}o$ = is the observed mean distance between each point and its nearest neighbor.

Denoted as:(2)$$\bar{D}o\;=\;\frac{\sum_{i=i}^{n}\;d_{i}}{n}$$
where; $\bar{D}a$ = is the expected mean distance for the points in a random distribution pattern.
(3)$$\bar{D}a\;=\;0.5\;(A/N)$$
where A is the minimum surface (square meters) that encloses a rectangle around all the points and N is the number of points. In general terms, the NNI is the ratio of the nearest neighbors’ distance observed between the random mean distances:NNI = d(observed)/d(random)(4)

If the result is higher than 1, the pattern is dispersed, if the result is lower than 1, the pattern denotes agglomeration. If the result is closer to 0, there will be a large concentration in the cloud of points as seen in Figure 1.

#### 3.2.2. Nearest Neighbor Hierarchical Clustering (NNHC)

Although NNI is a technique that helps determine if a distribution of points is dispersed or agglomerated, it does not identify the location of the clusters. Therefore, the NNHC technique is the second technique used to identify agglomerations of CLP points. This technique identifies groups of points that are spatially close [44], as shown in Figure 2.

This first spatial agglomeration generates the first-order clusters. Then another analysis is performed on the unusually close agglomeration and produced the second order of clustering. This analysis can be prolonged until there is no more associations. Usually, this analysis is limited to third-order clusters. For the cluster identification, the selected setting was five CLP cases or more with significant space agglomeration.

#### 3.2.3. Interpolation by Inverse Distance Weighted (IDW), Empirical Bayesian Kriging (EBK) and Kernel Density Approaches

The research addressed to analyze the spatial distribution of pollutants over a continuous space is the inverse distance weighted (IDW) interpolation, which assumes that things that are close to others are more similar than others that are far away. In order to predict a value in the space, IDW takes as reference its closest neighbors in a certain radius because neighbors who are closer to the point to be predicted will have more influence than the remote ones [44].

According to Cañada [45], spatial interpolation by IDW is denoted as follows:(5)$$Z(s_{0})\;=\;\sum_{j=i}^{n}\;\lambda \;\times \;Z(s_{i})$$
where Z($s_{0}$) is the value that predict the location (*s*_0_), *n* is the total sample points (emitting industry locations) near the point to be predicted, λ is the weighted value assigned to each point and it will be used for the prediction of values. The point values diminish with the distance, were Z($s_{i}$) is the value observed in the location $s_{i}$. In other words, the sample points that are further away from the point to be predicted within a given radius will have less weight with respect to those that are closer.

In addition, software sets as default a maximum of 15 nearest neighbor points and a minimum of 10. The weighted point values might have other coexisting weights. For PM_10_ values, the interpolation technique used was empirical Bayesian kriging (EBK), by means of the Geostatistical Analyst of ArcGIS 10.4.1, because this technique allows a better adjustment of air pollution data over a continuous space. With similar notation to Formula (5), the results were generalized by calculating the mean square error (MSE) denoted as follows:(6)$$MSE\;=\;\sqrt{\sum_{i=i}^{n}\frac{\hat{Z}\;\left(S_{i\;}\right)-\;Z\;{\left(S_{i}\right)}^{2}}{n}}$$
where $\hat{Z}\;\left(S_{i\;}\right)$ is the value after the interpolation and $Z\;\left(S_{i}\right)$ is the value measured at the point $S_{i\;}$. Similarly, Kernel Density was used to identify the areas of the MMA where CLP cases are intensified, which according to Kelsall & Diggle (1995) [46] is denoted as follows:
(7)$$g(xj)\;=\;\sum_{i=1}^{N}\left[KW_{i}I_{i}\frac{1}{h^{2}2\pi }e^{-\frac{d_{ij}^{2}}{2h^{2}}}\right]$$
where *g*(x*j*) is the density of cell *j*, $d_{ij}^{2}$ is the distance between cell *j* and a location of a CLP case *i*, *h* is the standard deviation of the normal distribution, K is a constant, $W_{i}$ is a weight in the location of a CLP case and $I_{i}$ is an intensity of the location of a CLP case.

#### 3.2.4. Spatial Scan Statistics (SaTScan)

Another form to detecting spatial clusters is the spatial analysis of Kulldorf. In this analysis the reference is not the distance between points (NNI analysis), but the population at risk in a particular area [47]. In this technique, we used the AGEBS with it population and CLP points. Finally, we compared the results of NNI and spatial scan statistics.

The SaTScan software has been used for health monitoring and to explore Clusters of disease in space, in time and space-time for congenital malformations [48,49,50]. The SaTScan use a Poisson model for discrete sample. This method permits to identify high risk groups in AGEBs associated with CLP. The expected number of cases in each AGEB is calculated as:(8)E[c]=p×C/P
where *c* is the observed number of CLP, *p* is the population of the census section of interest (AGEBS), and *C* and *P* are the total number of CLP and population, respectively. A relative risk of CLP for each AGEB is calculated by dividing the observed number of CLP by the expected number of CLP. The alternative hypothesis is that there is a high risk of CLP within the exploration window compared to the outside. Under the Poisson assumption, the likelihood function for a specific window is proportional to:(9)$${\left(\frac{c}{E\left[c\right]}\right)}^{e}{\left(\frac{C-c}{C-E\left[c\right]}\right)}^{c-e}I\left(*\right)\;$$
where *C* is the total number of CLPs, *c* is the observed number of CLPs within the window, and *E* [c] is the expected number of CLPs within the window under the null hypothesis that there is no difference. Because the analysis is conditioned to the fact that the total number of cases observed, *C* − *E* [*c*], is the expected number of cases outside the window. I(∗) is an indicator function, with I(∗) = 1, it is when the window has more cases than expected under the null hypothesis and 0 otherwise. The hypothesis test was carried out using 999 Monte Carlo simulations and of which a test statistic is calculated for each random repetition, as well as for the set of real data.

Log likelihood ratio (LLR) was calculated based on the difference of the incidences inside and outside the windows, and a Monte Carlo test helped to determine the statistical significance of the identified groups. A scan window with maximum LLR was considered the cluster with the highest probability, indicating that it was less likely to have happened by chance.

## 4. Results

### 4.1. Identification of Hierarchical Clusters for CLP and Emitting Industry

The spatial distribution of CLP cases, using a Kernel density analysis, showed a marked concentration in the periphery of the urban territory and the resulting clusters (Figure 3a,b), especially in the north and east of the MMA territory. Density and clustering analysis showed similar results.

The NNHC analysis produced 20 first-order clusters that agglomerated at least 5 CLP cases and 4 s order Clusters. This distribution is not a random distribution, but agglomerated in clusters. Figure 4 shows the spatial association between CLP cases grouped in clusters and the emitting industry Clusters.

In Figure 4, the first and second-order clusters can be observed. The second-order clusters are identified by numbers. The NNHC analysis formed 23 first-order clusters for industry and 4 second-order clusters. The second-order clusters for industry overlapped with second-order clusters for CLP in all, but the first. This last one is not overlaid with the second-order clusters of CLP, but it overlaps with 3 first-order clusters of industry. These methods showed the association between spatial concentration of cases with CLP and emitting industries, as shown in Table 3.

Table 3 shows the distribution of CLP cases, the total accumulation of substances and the amount of each one. In the second-order cluster 1 there are 22 cases of CLP that coexist with many substances including cadmium, methyl chloride and carbon dioxide. These substances had been linked to several adverse health effects [51]. In Cluster 3, there were 41 CLP cases that coexist with several pollutants: cyanides, cadmium, arsenate, mercury, lead, and carbon dioxide. The second-order cluster number 2 concentrate 61 cases of CLP, corresponding to the cluster with higher population and less pollutant concentration.

### 4.2. Nearest Neighbor Analysis

The agglomerations seen in Table 4 showed that both CLP cases and emitting industries tend to concentrate.

Table 4 shows the two main analysis units, CLP cases and polluting firms. The companies are disaggregated by types of polluting substances. Greenhouse gases are the main emission of industry, represented by 228 firms, while the cyanide industry is represented by only two firms. The NNI (column 6) shows that CLP cases and emitting companies have significant values of agglomeration with coefficients of 0.4 while aromatic, cyanide, and organic halogenated industries show random distributions (values > 1.0). It can be observed that companies have fewer points in space (301) than CLP cases (333); but companies have more Clusters than CLP. These is due to the neighboring of industries in the urban space.

### 4.3. Spatial Interpolation by Pollutant

Figure 5 shows the spatial location of emission industries with intensity gradients that can be associated with CLP cases.

Medium and high levels of pollutant concentration can be seen in the peripheric areas, as well as the CLP cases. Only aromatic pollutants tend to locate more on the center of the urban area. Table 4 shows the specific pollutant ranges.

In Table 5, CLP cases seem to be associated to different range of pollutant concentrations. For example, 84% of the CLP cases are located within areas of median and high concentration of gases (ranges 5 to 10). Most of the CLP cases are in range 5, representing 48.65% of the cases. About 96% of CLP cases are exposed to median and high values of aromatic pollutants and the pollutant range 5 accumulates 107 CLP cases (32.13%).

Organic halogenated pollutants followed a similar trend, where CLP cases are associated with higher pollutant concentrations. For metals and metalloids, most of the CLP cases are associated with low or median concentration ranges (ranges 1 to 5). For cyanides, most of the CLP cases were associated with the high-range levels (ranges 5 to 10) and 81 cases were observed in range 9 (24%). It is well known that cyanides are linked to congenital malformations [52]. PM_10_ exposure has also been linked to cardiovascular malformations [5,28,53]. In Figure 6, the CLP clusters of first and second order coexists with high PM_10_ concentrations.

The second-order clusters for CLP are identified by numbers; the higher PM_10_ values are in red and low values are in blue. This figure shows that second-order clusters of CLP 1 and 2 have clear interaction with high values of PM_10_; that is, significant grouping of CLP are associated with high values of environmental pollution, an association that could have implications in the detonation of certain health problems and particularly of congenital illnesses.

In Table 6, the CLP cases are distributed along their corresponding PM_10_ concentrations.

It can be seen that 68% of the cases are located in median and high values of PM_10_, with rank 8 being the highest number of cases concentrated with 58 cases (17.42%). It can be determined that most of the CLP cases present spatial interactions with high levels of pollution by PM_10_. Similarly, CLP spatial clusters are located in areas of the city with high levels of this type of environmental contamination. The concentration of the CLP cases is shown in Figure 7.

Figure 8 shows the association of cases with medium range-concentration of greenhouse gases, aromatics, and cyanides with PM_10_, probably with synergic mechanisms.

The principal components analysis (Figure 8) shows that greenhouse gases, aromatics, and PM_10_ in ranges between 4 and 7 coincide with the highest concentration of CLP cases, with an explanation of the variance of 53%. That means that the CLP cases present a greater interaction with this group of contaminants, a situation that could suggest that this group of pollutants could have greater influence on the incidence of this congenital disease.

### 4.4. Cluster Identification by Spatial Statistical Scan

Figure 9 shows the CLP clusters produced by two techniques: NNHC first and second-order clusters and SaTScan clusters based on AGEBS as the analysis unit. Secondary clusters 2, 3 and 5 (red circles) showed associations with second-order clusters 1, 2 and 4 (black circles). The most likely cluster (red circle) associates with the first-order clusters of CLP (filled in orange).

Particular attention generates the most likely Cluster, that is, the Cluster with the highest risk and probability of containing CLP based on the underlying population of the AGEBS. This cluster is located in the municipality of Apodaca (north of the city), a municipality that has a large number of industrial parks with highly polluting economic activities. These environmental agents have spatial interaction with clusters of CLP, a condition that could be intervening with some health problems.

The information of the five Clusters detected by the spatial statistical scan can be analyzed in Table 7, which shows that the most likely cluster concentrates 7 cases of CLP in a radius of 550 m, agglomerating 5 AGEBS with a population of 10,326 inhabitants. It can also be observed that this is the cluster with the highest relative risk, at 8.41; meaning that this Cluster has 8 times higher rate of CLP than observed outside. This is generated by a randomization process with levels of high significance.

Figure 9 shows that the most likely Cluster presents a strong spatial association with a first-order cluster generated by the NNHC technique. In this sense, if the CLP cases located within that cluster of the first order are added with the 7 located within the most likely cluster; these will group 12 cases in a radius of 1.2 km, a relatively short distance for a considerable number of cases of a congenital malformation. Similar reading can be given to the secondary cluster 4 located in the same municipality, which presents the greatest relative risk after the most likely cluster. This cluster has 5 cases of CLP in a radius of 420 m; however, it is not significant.

## 5. Discussion

We conducted an ecological exploratory research in order to identify distribution patterns and spatial associations of CLP cases with environmental pollutants. We found that CLP cases do not present a random distribution, measured with NNI and NNHC techniques. The results of these techniques showed several spatial coincidences, confirming the findings. The geographical or spatial identification of health conditions may be very useful in the understanding of the disease and for the creation of public policy.

CLP etiology probably does not come from a single factor, as a particular polluted ambient, but more likely from a multi-causality model. The ability to find particular geographical areas of health events may be useful to understand the ultimate cause or causes of disease and to use public policy to modulate or prevent it. CLP spatial distribution followed specific patters in the urban space. This coincides with the spatial agglomeration theory in the sense that everything is related to each other in space [54]. The present results do not establish a direct causality, but they indicate geographical proximity between CLP cases and ambient pollutants, as mentioned by other authors [55,56].

Several clusters of CLP cases were associated with carbon dioxide in first- and second-order clusters detected with NNHC. In addition, Clusters 1 and 2 showed an association with modest amounts of nickel, lead, cadmium, mercury, arsenic and cyanides. All these pollutants are linked to congenital malformations.

Even given that there is a lot of knowledge of genetic mechanisms in congenital malformations like CLP, many genetic interactions with ambient pollutions are not well understood, probably because multifactorial effects produce them [57]. Although more than 300 genes had been associated with CLP, recent research associate CLP cases with two candidate chromosomes, 17q and 11 [58,59], but the gene–environment interaction is still ambiguous [60,61]. More research is needed to understand this association. In order to study the influence of the ambient pollutants on CLP, personal exposure measures and characterization of pollutant point emissions are required. There are many variables that need to be adjusted, like tobacco or alcohol habits, socioeconomic level, medical access, vitamin consumption, and season of the year, among others [28].

There is a spatial association of CLP cases with PM_10_ concentrations. The CLP second-order clusters showed a clear interaction with high PM_10_ levels, so the significant grouping of CLP was associated with high values of environmental pollution. This association might explain the detonation of certain health problems, especially congenital malformations. The PM_10_ concentration varies along the year, showing high pics (over 100 µg/dL) several times.

### Limitations

For the identifications of points in the continuous space, it is necessary to have a precise location and this is not always easy to achieve. The environmental pollutants exposition were approximated to the addresses of the cases, but do not reflect the mobility of pregnant women to other locations.

In particular, this research faced the difficulty of the availability of information, because in Mexico there are no open data available on health problems of the population, particularly of congenital malformations, at a disaggregated level. The difficulty in conducting studies on a continuous space is increased by the confidentiality criteria that characterize this type of data, since performing a pattern analysis of points involves accurately identifying the location of the same. In this sense, it is clear that the data used to carry out this investigation constitutes a limitation, since data were not provided by a government agency, but by a civil association that deals with this type of congenital malformation.

## 6. Conclusions

Our study reviewed the spatial distribution of children with CLP and its association to environmental pollutants in MMA, which is one of the most polluted regions in Latin America. Although this study did not establish causal relationships, it showed the spatial interaction between CLP and environmental pollutants. With spatial statistical techniques, the space was treated as a continuous space and the CLP cases were individual points.

We acknowledge that the methods used for the present analysis have some limitation due to the quality of the data sources. We used a non-profit organization (Casa Azul A.C) data instead of public health data. Although we identify clusters of CLP, more precise data and statistics are needed to establish causality. This study provides a baseline description of air pollutions and CLP associations that will be important for future studies.

This research constitutes the first step to analyze the relationship between the CLP incidence from the spatial perspective with the use of open space applications. We have showed CLP agglomerations that interacted in space with different pollutants. More studies are needed to prove the interaction between the environment and the molecular biology of this disease. Our findings add important information to very few studies that have been published in Latin America.

## Figures and Tables

**Figure 1 ijerph-16-02488-f001:**
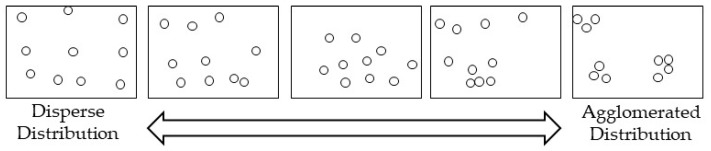
Type of spatial point distribution.

**Figure 2 ijerph-16-02488-f002:**
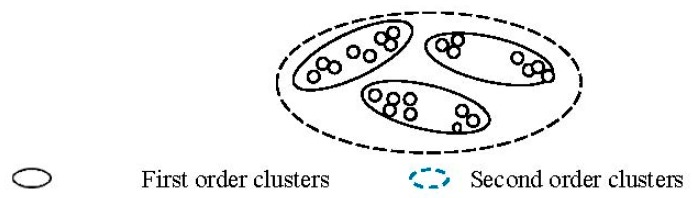
First and second order of cluster distribution.

**Figure 3 ijerph-16-02488-f003:**
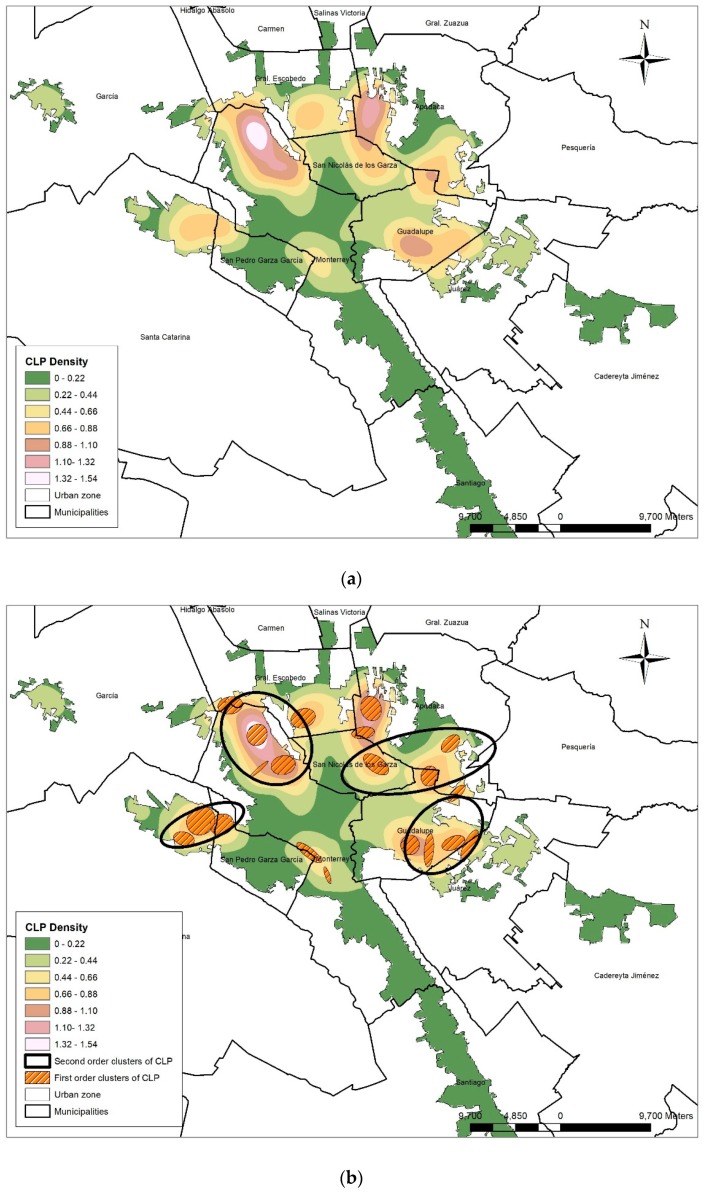
Spatial distribution of cleft lip and palate (CLP) in the Metropolitan Area of Monterrey. (**a**) Density; (**b**) Clusters.

**Figure 4 ijerph-16-02488-f004:**
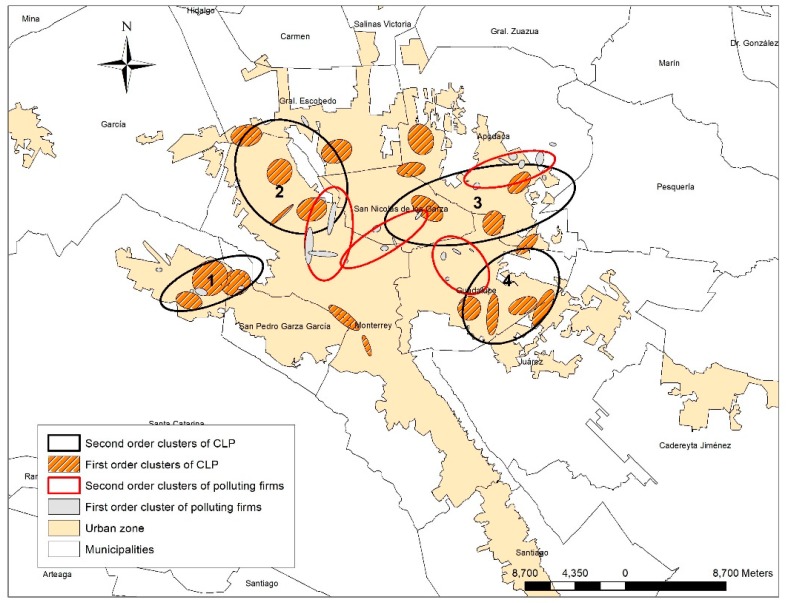
Spatial association of CLP and polluting industry.

**Figure 5 ijerph-16-02488-f005:**
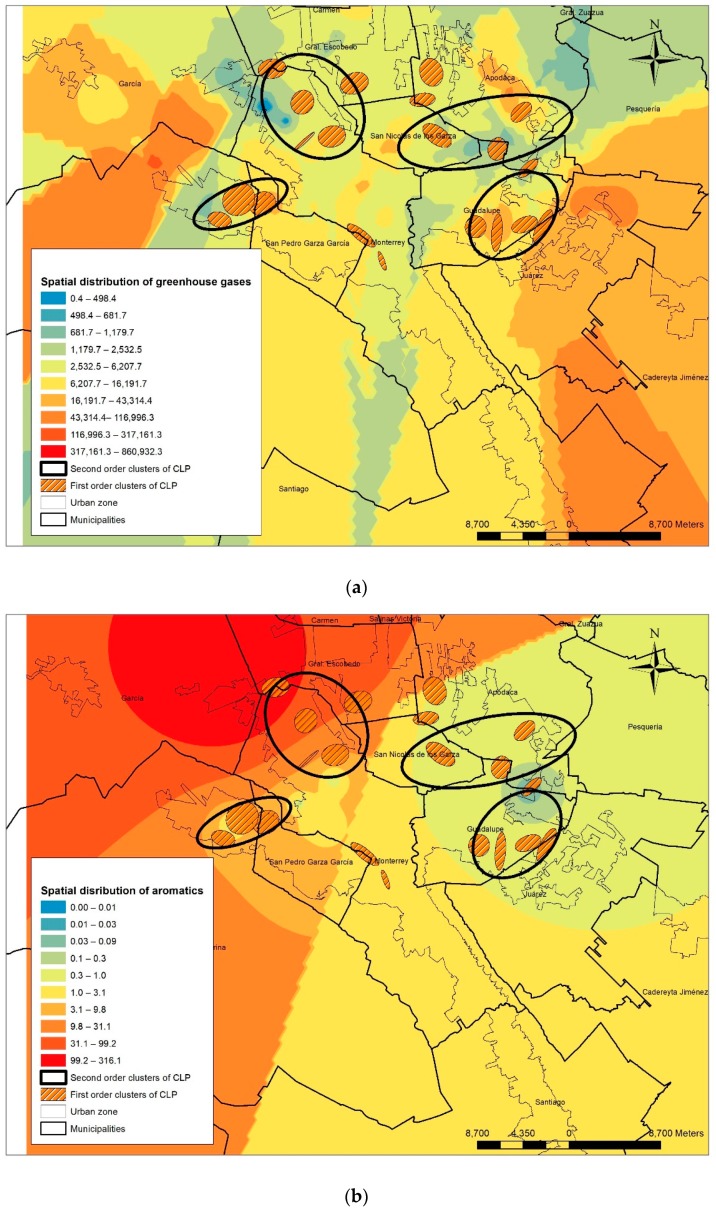
Spatial interpolation of industry´s pollutants. (**a**) Gases; (**b**) aromatic pollutants; (**c**) organic halogenated pollutants; (**d**) metals and metalloids; (**e**) cyanides and other pollutants.

**Figure 6 ijerph-16-02488-f006:**
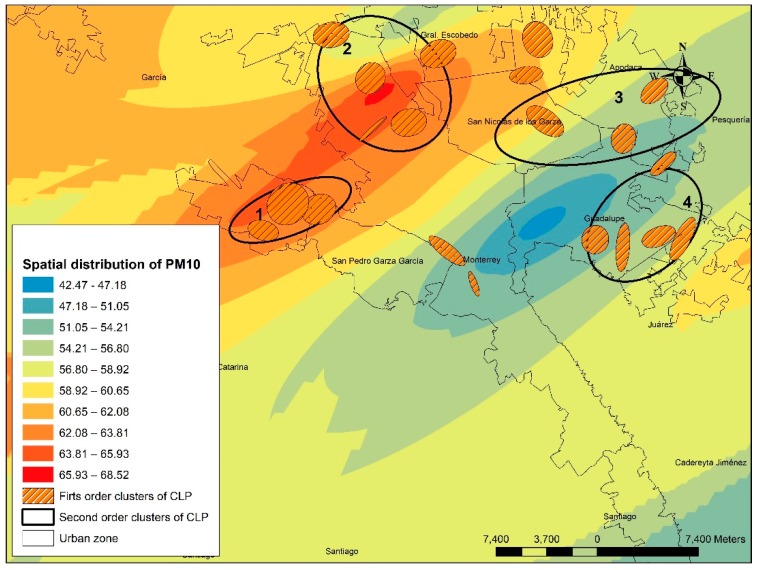
Spatial association of CLP cases and PM_10_.

**Figure 7 ijerph-16-02488-f007:**
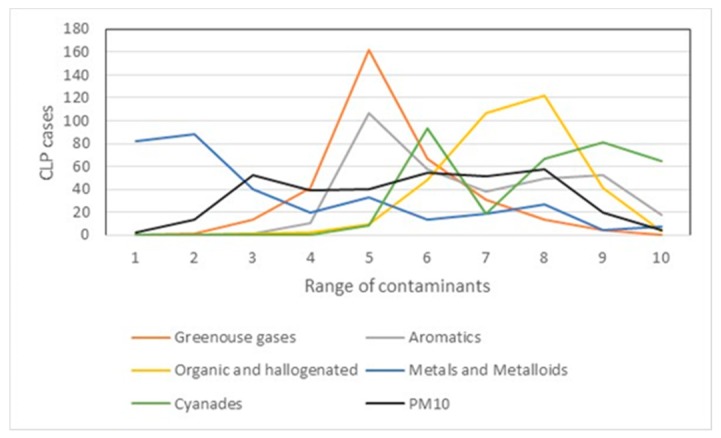
CLP cases by substance in different ranges of contamination.

**Figure 8 ijerph-16-02488-f008:**
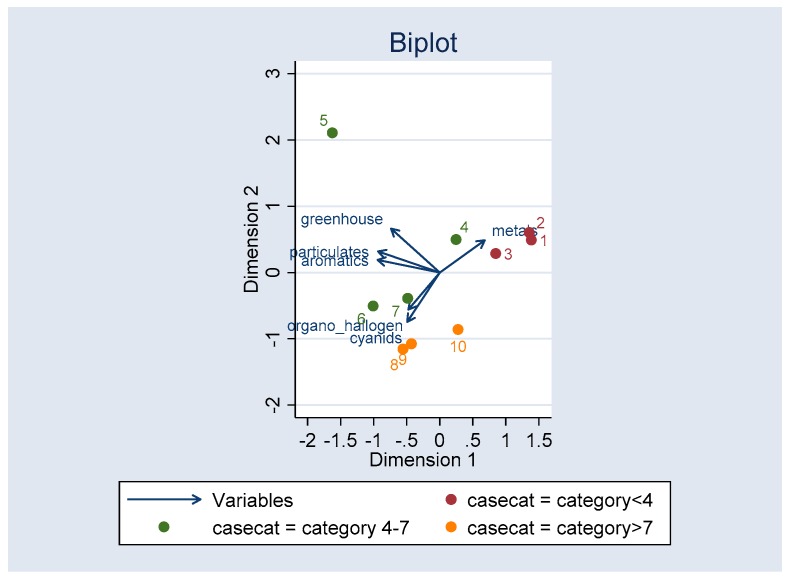
Clustering of cases by pollutant categories.

**Figure 9 ijerph-16-02488-f009:**
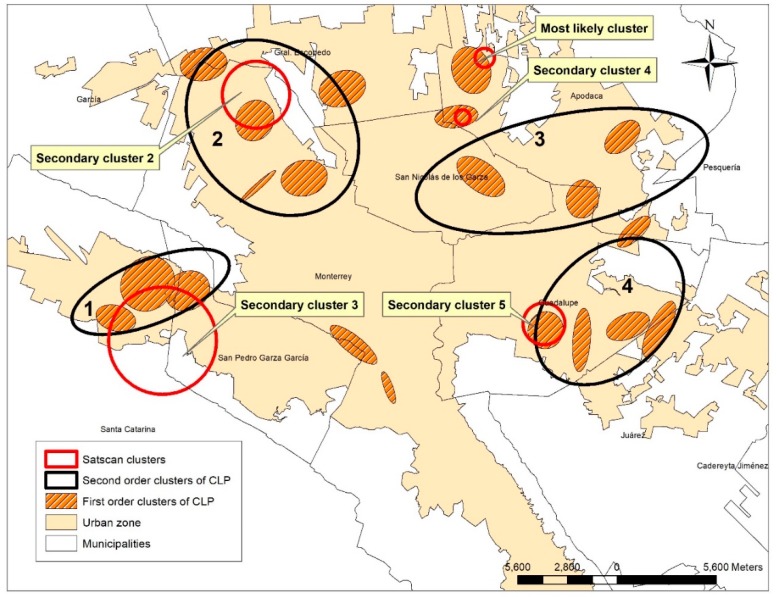
Cluster localization by SaTScan.

**Table 1 ijerph-16-02488-t001:** Classification of the main industrial pollutants. Emission sources.

**Cianides and Other Pollutants**	**Economic Activity**	**Contaminated Element**	**Anual Mean (tons)**
2-etoxyiethanol	Automotive manufacturing	Water	31.34
Polychlorinated Biphenyl	Energy poduction	Soil	
Cyanide inorganic/organic	Paints	Air	
dibutyl phthalate,	Home appliances		
Choride dioxide			
Sulfur formaldehyde hexafluoride			
**Aromatics Pollutants**	**Economic Activity**	**Contaminated Element**	**Anual Mean (tons)**
Styrene	Paints	Water	56.80
Phenol	Laminates	Air	
toluen diisocianate	Wholesale business		
**Halogenated Organic Pollutants**	**Economic Activity**	**Contaminated Element**	**Anual Mean (tons)**
1,2-dichlorobenzene	Food and beverages production	Water	12.8
1,4-dichlorobenzene	Chemical products productcion	Air	
chlorodifluoromethane	Refrigaration equipment		
Chloromethane			
methylene chloride			
hydrobromofluorocarbons			
**Metals**	**Economic Activity**	**Contaminated Element**	**Anual Mean (tons)**
Arsenic	Detergents	Water	794.4
Cadmium	Paper and cardboard production	Air	
Mercury	Car Motor production		
Chromium,			
Nickel			
Lead			
**Greenhouse and Combustion Gases**	**Economic Activity**	**Contaminated Element**	**Anual Mean (tons)**
Carbon dioxide	Refrigaration equipment	Air	1,613,046.7
Nitrogen dioxide	Electric Machinery		
Methane	Industrial baking		
Nitrogen dioxide	Hospitals		

**Table 2 ijerph-16-02488-t002:** Distribution of particulate matter (PM_10_) concentration by monitoring station during 2016.

Station	Obs	Mean	Std.	Min	Max
Southeast	52	42.5	10.4	27.3	76.6
Northeast	52	61.9	19.4	29.7	107.6
Downtown	52	55.5	15.6	33.1	99.2
Northwest	52	68.8	20.0	34.4	130.9
Southwest	52	64.6	19.1	33.0	119.7
Northwest (2)	52	78.6	21.7	32.0	127.2
North	52	53.2	11.0	34.5	78.7
Northeast (2)	52	57.4	11.8	37.1	85.8
Southeast (2)	52	61.6	14.7	38.8	102.1
Southwest (2)	52	62.0	14.0	38.1	102.1

**Table 3 ijerph-16-02488-t003:** Pollutant by cleft lip and palate (CLP) clusters.

Second-Order Cluster 1		Second-Order Cluster 2		Second-Order Cluster 3		Second-Order Cluster 4	
Total population	204,404	Total population	728,000	Total population	599,333	Total population	355,150
CLP cases	22	CLP cases	61	CLP cases	41	CLP cases	35
Total accumulation	8734	Total accumulation	1641	Total acumulado	6765	Total accumulation	3154
Pollutant	Cummulative amount (ton)	Pollutant	Cummulative amount (ton)	Pollutant	Cummulative amount (ton)	Pollutant	Cummulative amount (ton)
Carbon dioxide	8204.22	Carbon dioxide	1638.88	Carbon dioxide	4507.55	Carbon dioxide	3153.75
Nitrogen dioxide	472.49	Nitrogen dioxide	2.49308	Nickel (conpounds)	2203.93	Nickel (conpounds)	0.002527
Methyl Chloride	48	Lead (compound)	0.08229	Lead (compound)	29.0575		
Cadmium (Compounds)	2.7047	Nickel (conpounds)	0.02595	Cadmium	12.5141		
Lead (compound))	2.03935			Cyanide Inorganic/organic	5.56455		
Niquel (conpounds)	1.92753			Cromium (compounds)	4.35304		
Cromium (Compounds)	1.28269			Chlorodifluorometane (HCFC-22)	0.844		
Metane	0.748			Arsenic	0.50355		
Nitrous oxide	0.716			Mercury	0.243564		
1,4-DIChlorobencene	0.086368			Mercury (compounds)	0.0908		
Disosciated Toluene	0.0012						
Cadmium	0.00119						
1,2-Diclorobencene	0.00066						

**Table 4 ijerph-16-02488-t004:** Nearest Neighbor Index (NNI) for CLP cases and emitting industry by type of pollutant.

Chemical Type	Sample	First-Order Clusters	Mean Nearest Neighbor Distance (m)	Expected Nearest Neighbor Distance (m)	Nearest Neighbor Index (NNI)	*p*
Total CLP cases	333	20	634.0	1611.9	0.4	0.001
Total Polluting Firms	301	23	438.8	1125.2	0.4	0.001
Firms emitting Greenhouse Gases	228	7	537.1	1292.8	0.4	0.001
Firms emitting Aromatics	5	0	3760.8	2379.2	1.6	0.965
Firms emitting Organic-Halogenated	13	0	3111.0	2861.5	1.1	0.857
Firms emitting Metals and metalloids	53	2	1434.9	2316.6	0.6	0.054
Firms emitting Cyanides and other pollutants	2	0	20,026.6	3505.8	5.7	0.988

*p*: Significance levels.

**Table ijerph-16-02488-t005a:** (**a**) Gases.

Range	Values	CLP Cases	Percentage
1	0.4–498.4	0	0.00
2	498.4–681.7	1	0.30
3	681.7–1,179.7	13	3.90
4	1179.7–2532.5	41	12.31
5	2532.5–6207.7	162	48.65
6	6207.7–16,191.7	67	20.12
7	16,191.7–43,314.4	31	9.31
8	43,314.4−116,996.3	14	4.20
9	116,996.3–317,161.3	4	1.20
10	317,161.3–860,932.3	0	0.00

**Table ijerph-16-02488-t005b:** (**b**) Aromatic pollutants.

Range	Values	CLP Cases	Percentage
1	0.00–0.01	0	0.00
2	0.01–0.03	0	0.00
3	0.03–0.09	1	0.30
4	0.1–0.3	10	3.00
5	0.3–1.0	107	32.13
6	1.0–3.1	58	17.42
7	3.1–9.8	38	11.41
8	9.8–31.1	49	14.71
9	31.1–99.2	52	15.62
10	99.2–316.1	18	5.41

**Table ijerph-16-02488-t005c:** (**c**) Organic-halogenated pollutants.

Range	Values	CLP Cases	Percentage
1	0.00–0.02	0	0.00
2	0.02–0.05	0	0.00
3	0.05–0.11	1	0.30
4	0.11–0.23	2	0.60
5	0.2–0.5	9	2.70
6	0.5–1.0	48	14.41
7	1.0–1.9	107	32.13
8	1.9–3.9	122	36.64
9	3.9–7.9	41	12.31
10	7.9–16.0	3	0.90

**Table ijerph-16-02488-t005d:** (**d**) Metals and metalloids.

Range	Values	CLP Cases	Percentage
1	0.00–0.01	82	24.62
2	0.01–0.04	88	26.43
3	0.04–0.16	40	12.01
4	0.16–0.6	20	6.01
5	0.6–1.9	33	9.91
6	1.9–6.4	13	3.90
7	6.4–21.5	19	5.71
8	21.5–72.9	27	8.11
9	72.9–246.6	4	1.20
10	246.6–834.6	7	2.10

**Table ijerph-16-02488-t005e:** (**e**) Cyanide and other pollutants.

Range	Values	CLP Cases	Percentage
1	0.001–0.003	0	0.00
2	0.003–0.01	0	0.00
3	0.01–0.04	0	0.00
4	0.04–0.12	0	0.00
5	0.12–0.39	8	2.40
6	0.39–1.3	93	27.93
7	1.3–4.1	19	5.71
8	4.1–13.5	67	20.12
9	13.5–43.8	81	24.32
10	43.8–142.4	65	19.52

Source: own elaboration.

**Table 6 ijerph-16-02488-t006:** CLP cases and PM_10_ concentration.

Range	PM_10_ Concentration (mg/dL)	CLP Cases	Percentage
1	42.47–47.18	2	0.60
2	47.18–51.05	13	3.90
3	51.05–54.21	52	15.62
4	54.21–56.80	39	11.71
5	56.80–58.92	40	12.01
6	58.92–60.65	54	16.22
7	60.65–62.08	51	15.32
8	62.08–63.81	58	17.42
9	63.81–65.93	20	6.01
10	65.93–68.52	44	1.20

**Table 7 ijerph-16-02488-t007:** Detected clusters by SaTScan.

Cluster	Latitude	Longitude	Radio (km)	Number of AGEBS	Population	Observed	Expected	RR	LLR	*p*-Value
Most likely cluster	25.79678583	−100.2428446	0.55	5	10,326	7	1.85	8.41	8.68	0.005
Secondary cluster 2	25.77927943	−100.3725095	1.90	26	102,750	21	8.44	2.59	6.82	0.001
Secondary cluster 3	25.65391682	−100.4269285	3.05	24	60,032	14	4.93	2.92	5.66	0.050
Secondary cluster 4	25.76653482	−100.2557045	0.42	2	10,015	5	1.82	6.15	4.87	0.098
Secondary cluster 5	25.66025114	−100.2111653	1.19	11	44,792	11	3.68	3.06	4.81	0.595

AGEBS: Geostatistical basic area; RR: Relative Risk; LLR: Likelihood ratio.

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
