# Peer review of "Spatial Clusters of Children with Cleft Lip and Palate and Their Association with Polluted Zones in the Monterrey Metropolitan Area"

_ijerph, 2019, doi:10.3390/ijerph16142488_

Round 1
Reviewer 1 Report
This paper is on a very interesting topic, and the authors clearly went through extensive effort to conduct their literature review, explain the statistical methodology and acquire data. However, there are some fundamental flaws in the way the data sources are used and places where more clarification is needed.
1.The fact that the data come from persons who are low income biases the spatial locations of cases. Persons from impoverished communities tend to live together. Therefore, the statistics used (such as the Poisson model) must at minimum be adjusted for the average percentage of the community in a certain zip code that lives in poverty. Otherwise, all you are doing is identifying high-poverty neighborhoods.
2. The air pollution data need to be more carefully explained. It is very unclear to me how they go from a list of pollutants to the IDW maps. It seems as though the list of industries is used to suggest levels of different pollutants, however there are very few actual numbers (PM10 being the exception). How do you move from the qualitative list of polluters to the quantitative maps?
3. IDW is not an appropriate statistic for many types of air pollution, because it assumes the same relationship across space. For example, PM10 is heavy and may fall out of the air more quickly, so that the concentrations are more localized than IDW indicates. However, ozone tends to be much more dispersed. Something like Empirical Bayesian Kriging (available with ArcGIS) would be more appropriate, if monitoring data are available for other pollutants, because it lets the data determine the spatial relationship and gives a measure of predicted error.
4. The first results shown are for kernel density, but kernel density is not mentioned by name prior to those figures.
5. Reduce the number of significant digits in the tables and figures. You are not measuring them at such a high level of specificity.
6. With the above being said, I do think this paper tackles a really interesting question, and the variety and extent of pollution in the area deserve attention, as does CLP. I think the data and analysis need to be rethought for it to truly suggest associations that can be followed up.
Author Response
Dear. Editor and Reviewers, we revised and corrected all the issues presented by the reviews. Please see our responses attached.
Augusto Rojas, MD, PhD

Reviewer 2 Report
As a reviewer I have the following remarks
Lines29-30, please express 1/700, 1/1500 with denominator 1000, as for Mexico.
Line 41 Define PM10 and 2.5.
Table 1: List “nitrogen dioxide” no NO2.
Table 2, values round to one digit: 42.46 show as 42.5.
Your formulas 1-4 show with /, say NND=Do/Da – now the fractions have small letter.
Line 83 Don’t use “next Figure” – which one? Say number.
Your formula (6) is OK (see my point 5).
In formula 7 I suggest to write I(*) – to show that it has an argument.
Figure 2, in the legend we have to may digits – round to two if possible. Distance bar could be large –we don’t see the scale.
Line 264 “carbon dioxide (CO2)”
Table 4 Reduce the number of digits – round. It’s difficult to read/understand.
Figure 5, as in #9.
Table 5- please round the number, do we need 11.41%?, better 41.4% or even 41%.
In general, clean the paper, round the presented numbers.
Thank you
Author Response

(The authors gave the same response as above.)

Reviewer 3 Report
This paper presents a spatial clusters analysis of Children with cleft lip and palate and their association with polluted zones in the Monterrey Metropolitan Area, which is an interesting topic, and of relevance to the International Journal of Environmental Research and Public Health journal. There were a couple of grammar errors and typos. More specific comments can be found in the following. Some areas where I would like to see more detail:
1) -- Will you draw a detailed diagram of the methods for your study?
2) Typos
3..Method should be 3.Method
ArcGis should be ArcGIS
4) Will you draw a map about the research data in Part 3?
5) How to choose the value of parameters in your model? Please give more information about the information (like the number of clusters).
6) Please use the right form for the references.
7) The conclusion part should be improved. Please discuss the uncertainty and accuracy of your methods.
8) Please use the normal font size in figures. Figure 1 should be updated with high quality.
Author Response

(The authors gave the same response as above.)
